# Introducing the Pi-CON Methodology to Overcome Usability Deficits during Remote Patient Monitoring

**DOI:** 10.3390/s24072260

**Published:** 2024-04-02

**Authors:** Steffen Baumann, Richard Stone, Joseph Yun-Ming Kim

**Affiliations:** Industrial and Manufacturing Systems Engineering, Iowa State University, 2529 Union Dr, Ames, IA 50011, USA; steffen.baumann@gmail.com (S.B.); rstone@iastate.edu (R.S.)

**Keywords:** remote patient monitoring, passive sensors, usability, Pi-CON methodology, virtual health, telehealth, IoT, gait monitoring, handsfree vital signs

## Abstract

The adoption of telehealth has soared, and with that the acceptance of Remote Patient Monitoring (RPM) and virtual care. A review of the literature illustrates, however, that poor device usability can impact the generated data when using Patient-Generated Health Data (PGHD) devices, such as wearables or home use medical devices, when used outside a health facility. The Pi-CON methodology is introduced to overcome these challenges and guide the definition of user-friendly and intuitive devices in the future. Pi-CON stands for passive, continuous, and non-contact, and describes the ability to acquire health data, such as vital signs, continuously and passively with limited user interaction and without attaching any sensors to the patient. The paper highlights the advantages of Pi-CON by leveraging various sensors and techniques, such as radar, remote photoplethysmography, and infrared. It illustrates potential concerns and discusses future applications Pi-CON could be used for, including gait and fall monitoring by installing an omnipresent sensor based on the Pi-CON methodology. This would allow automatic data collection once a person is recognized, and could be extended with an integrated gateway so multiple cameras could be installed to enable data feeds to a cloud-based interface, allowing clinicians and family members to monitor patient health status remotely at any time.

## 1. Introduction

Healthcare is moving from the hospital to the home [1] driven by an abundance of newly launched consumer health devices such as wearables, fitness trackers, or home use medical devices that allow collecting patient health data while the patient is in their home environment. These devices, also called Patient-Generated Health Data (PGHD) devices, collect patient data such as step count, heart rate, sleep quality, respiration rate, and blood pressure, have been shown to reduce cost, hospital readmissions, and the potential of hospital-associated infections [2]. This is possible with the novel ability for healthcare providers to access more comprehensive data between patient visits, which enhances individual care by better understanding the patient’s symptoms and progression of a health condition [3].

### 1.1. The Acceleration of Telehealth and Its Benefits

This trend contributes to the acceleration of telehealth, which is an umbrella term that integrates several health services enabled through innovative communication technologies, as shown in Figure 1. According to Catalyst [4], telehealth includes:The option for a patient and clinician to connect over a video feed;“Store and Forward”, a capability for specialty services (such as diagnostic imaging) to capture and transmit patient data (in this case, an X-ray) to another healthcare provider;mHealth, allowing communication and access to patient records via mobile devices;Remote Patient Monitoring (RPM).

**Figure 1 sensors-24-02260-f001:**
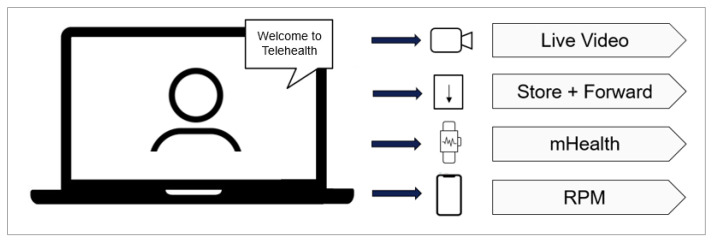
The subcomponents of telehealth.

RPM enables the capture of patient health data at the patient’s home, with the healthcare provider’s ability to review the data remotely and monitor the patient’s health condition. As these technologies evolve, they provide increased opportunities to further modernize and streamline healthcare delivery by embracing telehealth and its subsets.

Moving care away from the facility and close to the patient has many advantages, including decreasing readmissions and costs. The period following hospitalization can be the most critical time for a patient. In fact, as many as 20% will suffer adverse events after hospital discharge, which will cause hospital readmission [5]. RPM has demonstrated its potential to significantly contribute to reducing these hospital readmissions. Freitag et al. [6] report data from 2018 demonstrating that hospitals utilizing RPM show a 75% reduction in hospital readmissions. Very similar numbers are reported by Gjeka et al. [7], claiming a readmission rate reduction of 78%.

RPM has not only been demonstrated to reduce the number of hospital readmissions but is also advantageous for the reduction in cost and Hospital-Acquired Infections (HAI):Spelman [8] highlights that a report published by the Australian National Nosocomial Prevalence Survey stated 6.3% of HAI in 1984, and a report released by Magill et al. [9] confirmed this number from 1984, reporting that amongst 183 surveyed hospitals in 2011, 4% of patients suffered one or more HAI in the U.S.;Piccini et al. [10] report that for all facilities participating in RPM, costs are reduced by 30% per year (USD8720 versus USD12,423).

Another factor that shows the value of remotely monitoring a patient at their home is to combat the nursing shortage that takes its toll globally, as more elderly or postoperative patients seek medical attention, the patient-to-staff ratio will continue to widen and impact patient care. Since the onset of the COVID-19 pandemic, many nurses retired early or stopped working in their jobs and changed fields due to high-stress levels and burnout [11]. A reported 1118 hospitals, or 17%, declared a shortage in nurses [12], and many states issued temporary licenses to nursing students to address this shortage [13].

In addition to monitoring elderly and postoperative patients, RPM is also beneficial for monitoring a patient’s acute or chronic health condition and detecting patients’ potential falls in their homes [14]. This allows the clinician to draw conclusions about the patient’s current health, avoiding adverse patient events in the future. This is especially helpful for those living in remote areas. According to Ameh et al. [15], 60 million Americans live in rural areas and have limited access to care, impacting their health and overall well-being. Medicare and many insurance companies cover the visit of a home care nurse, but this is only for home-bound patients [16], so many patients in rural areas would still have to make a trip to their healthcare provider, which can cause hardships to many.

In this paper, the difficulties and disadvantages of current PGHD devices and its data generation are discussed and categorized into four significant challenges: patient’s characteristics, technical device limitations, patient compliance to RPM, and device placement on body. Each of these issues impacts current devices’ accuracy, usability, and efficiency and poses significant risks to allowing for the full adoption of PGHD devices. After discussing these current limitations and risks, the Passive, Continuous and Non-Contact (Pi-CON) methodology, a novel methodology for gathering patient-generated data, is proposed. This framework works to avert each of the primary issues caused by current RPM technology, while simultaneously being just as accurate and more efficient due to continued data recording.

### 1.2. Background

Despite the many upsides of home use medical equipment in improving patient care, the accuracy and validity of these physiological metrics is a significant cause of concern among patients [17] and clinicians [18]. In a study performed in 2009, for example, the accuracy of 554 automated blood pressure devices was studied, with only 30% of the devices showing acceptable accuracy [19].

The inception and expansion of PGHD devices in recent years have made this a critical period in developing novel technologies. During this transformation period involving remotely monitoring the patient’s health and empowering the patient to take health exams by themselves, proper usability of these devices and interaction across all included stakeholders, including healthcare providers, patients, and insurance companies [20] is crucial to ensure proper future patient care.

In Figure 2, the RPM Usability Impact Model describes the unique challenges that impact a PGHD device’s usability. The four major pillars of this model are Patient’s Characteristics, Technical Device Limitations, Patient Compliance to RPM, and Device Placement on Body. Each of those will be explained further below.

### 1.3. Patient Characteristics

An advantage and challenge of healthcare is the unique physiology and anthropometry of every person due to the complex nature of the human body. As a result, it is essential to consider a multitude of patient characteristics to ensure that usability is inclusive for all people, regardless of any physical or psychiatric qualities they might possess. Inclusivity in the case of usability means these devices are incredibly intuitive to learn and use while simultaneously inspiring confidence in the data’s accuracy.

One of the most critical groups to consider is patients with physical disabilities, chronic pain (such as musculoskeletal conditions), and/or chronic illnesses. These patients’ health conditions may prevent them from participating in RPM or virtual health programs if they suffer from complications, such as respiratory issues or immobility. Per Bowles et al., 24% of patients refused to use a PGHD device because they were “too sick to bother” [21]. Additionally, if devices are seen to do more harm than good, such as a blood pressure cuff squeezing too tight on someone with chronic pain in their arms, it can be detrimental to the technology’s usability. Considering this group is one of the groups most in need of medical attention, it would be unethical to show indifference, much less exclude, these patients that could benefit immensely from improvements to this technology’s usability.

In addition to considering how physical characteristics may negatively impact the way patients interact with PGHD devices, designing technology for patients with psychiatric disabilities or conditions, such as anxiety disorders, depression, or autism spectrum disorders is necessary [22]. Rodríguez-Fernández et al. report that around 13 million adults in the US may be challenged to adopt a Telehealth service due to difficulties [23], and Vázquez-de Sebastián et al. add that a patient’s sensory limitations may hinder the use in mobile health technologies [24]. What can cause this lack of adherence is that these devices may not have been designed with these users in mind.

Regardless of physical or psychological characteristics, there are individual attributes that affect usability. This starts with the ability to understand health data results and according to Cutilli and Bennett [25], 14% of U.S. adults are health literate at a below-basic level.

Chaniaud et al. [26] investigated the effects of limited device usability based on the participant’s age, education level, technical knowledge, and health literacy, with the results revealing a positive relationship between age, health literacy, and usability, showing the importance of obtaining the necessary health knowledge to understand the results to track the health status. The authors also tested device usability through direct observation and questionnaires, learning that users with high levels of health literacy were more comfortable using the device.

In another study, Reyes et al., reported [27] the results of a usability study with a home use multi-parameter monitor (a device that measures blood pressure, heart rate, SpO2, respiration rate, and body temperature) to determine whether there were shortcomings on the device based on limited usability. Results showed that 92.3% of all usability problems were related to users being unfamiliar with medical abbreviations, leading to incomprehension of measurements. Additionally, 76.6% of the study subjects were unable to interpret the results on the monitor, which is necessary to stay motivated during RPM.

### 1.4. Technical Device Limitations

Traditional home use medical devices often have multiple components that can malfunction. Batteries can fail, sensors can become loose or misaligned, and wearable straps can break. These breakdowns can disrupt data collection and require device replacement or repair. Additionally, build-up of dirt, sweat, or other debris on the sensors can interfere with their ability to collect accurate data.

Additionally, displays on home medical devices might malfunction, making it difficult or impossible for patients to interact with the device, which is necessary for these types of instruments. While at healthcare facilities medical professionals perform procedures, such as recording vital signs or spirometer tests, at home, the patient must rely on the training received from the medical professional and manage exams independently. If the device completely stops working, data will not be transferred. Additionally, if a device requires calibration or no longer works accurately, it could lead to inaccurate patient data.

### 1.5. Patient Compliance to RPM

Improper device function, such as a battery that does not hold an adequate charge, can contribute to limited patient adherence to an RPM program. When a patient is not compliant and will not engage in the program as requested by the healthcare provider, insufficient data will be collected, potentially impacting the treatment plan. The patient may start motivated, but compliance with the examination schedule may drop, as also demonstrated by the Novelty Effect. The Novelty Effect is defined as the “tendency for performance to initially improve when new technology is instituted, not because of any actual improvement in learning or achievement, but in response to increased interest in the new technology [28]”. Shin [29] demonstrates this by reporting that most Fitbit users stopped using the device after three months. And with the shift to value-based care, a new type of compensation system has been introduced to reimburse providers based on the patient’s health outcome rather than on the tests ordered [30]. Therefore, providers must rely on the patient’s adherence to using and wearing the device regularly and properly as instructed.

In an experimental study, Michaud et al. [31] reported that only 61% of participants complied to a remote patient monitoring treatment protocol for taking their own vital signs for two weeks twice a day, and Motolese et al. [32] discovered in their experiment an RPM-participating compliance rate of approximately 27%, with demotivation, technical problems, and digital illiteracy as the main factors.

In order to ensure continued engagement by the patient and to provide maximum usability, the cost of interaction during device operation needs to be kept to a minimum, without the need for the user to remember the steps necessary [33]. Additionally, many health conditions, for instance, require nocturnal monitoring, such as hypertension [34], making human interaction not only impossible but also something the patient has to remember and potentially disturbing a good-night rest.

Per Hossain [35], the cost of interaction is comprised of the following elements that make up the effort to carry out a task, such as a medical exam:Interaction: requires action by the user to get an exam started;Attention: demands for the user to focus that all necessary steps are carried out;Thinking: necessary means to interact with a device;Reading: required if thinking alone does not achieve the goal.

COVID-19 has contributed to an increased interest in non-contact patient monitoring technologies due to the eliminated risk it provides of spreading the virus [36]. Devices to monitor patients should be designed to not only limit the patients’ contact with the device but the device should also be small, flexible, hypoallergenic, and not bother patients during their daily activities [37,38,39].

In addition to the risk of infection, other factors that impact the patient’s adherence to wearing a device is the importance of visual appearance when the device is worn. Aesthetics was a significant factor influencing both perceived enjoyment and social image of wearing the device, according to the research by Hsiao and Chen [40] and Yang et al. [41]. Another aspect is the fact that many people dislike having something attached to their bodies. Sticky adhesives or bulky smart clothing may not be acceptable in all patient populations due to general discomfort or allergic skin reactions. Berg et al. [42] report that 35% of continuous glucose monitoring users developed a minimum of one skin lesion due to the worn devices, and similarly, Wong et al. [43] report their findings of a study where nearly half of all users (727 of 1662) stopped wearing a glucose monitoring sensor after one year due to discomfort (42%) and developed skin reactions caused by the device as a major complaint. Additionally, in a study by Jeffs et al., 208 patients were equipped with a wearable monitor after discharge from an intensive care unit to monitor physiological deterioration [44]. Among the 192 participants included in the analysis, 32% removed the monitor before the study finished due to discomfort.

### 1.6. Device Placement on Body

There have been a variety of usability challenges reported when the patient or the patient’s caregiver is responsible for the device’s operation at their home [26,45,46]. Muntner et al. [47] reported, for instance, that the American Medical Association (AMA) advises clinicians to keep the patient’s back supported, as well as keeping their legs still by having both feet on the floor during a blood pressure measurement. The authors add that not complying with this guideline could change a blood pressure reading by as much as 15 mmHg for a systolic reading. Murakami and Rakotz [48] published very similar data, demonstrating that systolic blood pressure readings could vary as high as 40 mmHg when a blood pressure cuff is placed over clothing on the arm, or up to 15 mmHg when the patient has a full bladder during the exam or if engaged in a conversation during the exam.

### 1.7. Necessity for Continuous Data

In current PRM programs, healthcare facilities provide the patient with a remote patient monitoring kit that includes a set of devices and requires the patients themselves to monitor their health condition on average three times per day, depending on the physician’s treatment plan [49]. This RPM kit usually consists of devices such as a pulse oximeter, thermometer, and scale so the patient can carry out spot-checks throughout the day [50]. Devices that continuously collect and monitor patient data, however, can be better indicators of a patient’s actual health rather than just a snapshot, especially when said snapshots can be inaccurate from factors such as white coat syndrome (increased blood pressure due to physically being in a healthcare setting [51]).

Another challenge with monitoring patients remotely is that data capture may not only be inaccurate, but captured data may be incomplete or limited only to Detect Atrial ibrillation (AFib), for instance, a condition that causes irregular heartbeat [52], an abnormal heartbeat may be present during a time when the heart rate is not monitored by the user [53].

Continuous health data will enable faster interventions if health issues arise, allowing the physician to get more data beyond the patient’s spot checks. According to O’Malley [54], approximately 90% of postoperative patients in a hospital setting are spot-checked daily, meaning a clinician checks the patient’s health status and vital signs several hours apart. As a result of the intermittent nature of spot checks, the caregiver may miss a significant vital sign event that typically precedes severe adverse events, and Michard and Kalkman [55] assert that up to 90% of below-normal oxygen levels and 50% of abnormally low blood pressure readings are missed when spot-checking a patient. Continuous patient data also allows health practitioners to better analyze and detect new opportunities for effective intervention. In the U.S. alone, over 400,000 patients die in hospitals every year from preventable adverse events [56], and Brown et al. state that a lack of continuous monitoring is a contributing factor to in-hospital mortality, disclosing that medical emergencies are reduced by 86% when the patient is continuously monitored [53].

## 2. The Pi-CON Methodology

The aforementioned examples solicit a solution that removes the patient’s burden of proper data acquisition and compliance to automate data generation with limited user interaction and reporting health data continuously.

One of the main benefits of RPM is the early detection of patient deterioration [57], for which vital signs are representative indicators [58]. Per the John Hopkins Hospital [59], the three main vital signs are body temperature, heart rate, and respiratory rate. Among these three, heart rate and respiratory rate are critical and at the same time leading indicators to accurately and timely detect a patient’s declining health condition [60]. Indeed, Philip et al. state that respiratory rate is known as “the most sensitive marker of a deteriorating patient” [61], and Elliott adds that respiratory rate “is the most neglected vital sign” [62]. In a clinical setting, respiratory rate is difficult to examine and is usually determined by manually counting chest-wall movements [63]. However, this is prone to errors, and just as with other vital signs that are spot checked only, it is not feasible to capture respiratory rate continuously.

When considering the importance of continuous and at-home health monitoring, including its challenges, novel technologies that are equally usable and minimally invasive must be researched and integrated. To solve these issues, we propose utilizing the Pi-CON methodology. Pi-CON stands for Passive, Continuous and Non-contact and describes a data collection method that eliminates the potential of human error by keeping the cost of human interaction to a minimum and integrates continuous data generation without attaching any devices or sensors to the patient’s body. For this, an omnipresent sensor shall be used to determine a patient’s health condition ubiquitously, avoiding inaccurate patient data acquisition by inexperienced users in an “unregulated” care environment [64]. This methodology works to prevent the usability issues reported and that patients are experiencing daily when they are asked to put their health in their own hands, taking medical examinations without the presence of a medical professional. To get all benefits of Pi-CON, a fully integrated sensor setup should be cloud-based, or Internet of Things (IoT)-enabled, so that the data captured can be viewed in near-real time in a cloud-based portal that’s accessible from anywhere at any time. Since Pi-CON is proposing to collect data continuously, the real value is achieved when this continuously tracked health data can actually be reviewed as needed to determine a full history and health trends based physical activity and lifestyle.

### 2.1. Recommended Technology to Apply the Pi-CON Methodology

In order to implement Pi-CON, there are multiple technologies that may be suitable for remotely and continuously monitoring patients. Each possible option will be discussed in detail in the following sections, including their advantages and disadvantages, with the objective to identify a technology that allows for accurate data generation and low complexity to ultimately allow high market adoption of Pi-CON.

#### 2.1.1. Radar

Radar (Radio Detection and Ranging) transmits and detects Radio Frequency (RF) waves. As shown in Figure 3, the technology works by transmitting short pulses from a radar module (Tx), followed by processing the signal with a receiver (Rx) once the signal has bounced off a target [65].

Radars are used not only to detect and track airplanes or get weather information but also to detect motions. Therefore, radar technology lends itself well for indoor monitoring. Due to its reliability, safety, and ability to penetrate through objects and walls, it suits itself well for remote patient monitoring [66]. Radar systems are often designated by the wavelength and frequency in which they operate and operate on several bandwidths, with some being ideal for patients’ health and motion monitoring [67]. There are different types of radars, such as Continuous Wave (CW) radar, Frequency Modulated Continuous Wave (FMCW) radar, or pulse radar, each with different properties and frequencies, ranging from 3 MHz–300 GHz, depending on its applications and type [68].

A special type of pulsed radar is Ultra-Wideband (UWB) radar. It has received recent attention due to its ability to physiological signals detection, human target positioning, and wireless communications. It utilizes a wide radar spectrum, as the name suggests, and has more penetrating capabilities through obstacles, providing excellent range due to the short pulse duration, leading to high precision in target detection [69]. Therefore, it is suited well for determining vital signs and estimating a human’s respiratory rate by determining the displacement of a chest and detecting the chest expansion every time a person inhales [70] (see Figure 4). In a relaxed subject, a chest displacement while breathing can induce displacements between 4 mm to 12 mm, depending on the person [69,71].

UWB radars operate between 3.1 GHz and 10.6 GHz [72], and FWCW at around 60 GHz [73]. Therefore, a higher frequency means less penetration through objects, so vital signs determination is often preferred with UWB due to its penetration properties and high bandwidth, which can also facilitate high data rates [74].

Both Google and Apple started taking advantage of these properties and are utilizing radar applications in their products. Apple has started to utilize UWB with the launch of the iPhone 11 to identify the direction and distance of an object and its path. It works in concert with other communication protocols (such as Bluetooth or Wi-Fi) integrated into the phone and utilized the fastest and best-suited protocol for the desired application [75]. Google is taking a different approach with Project Soli and is innovatively utilizing FMCW to integrate gesture-based interactions at close range into their devices. This technology’s intent is to change how consumers interact with their devices, such as phones or wearables, and even other home appliances. The objective is to eliminate conventional input devices, such as a computer mouse, touch screen, or keypad, and use hand gestures instead to interact with the device. Therefore, rather than touching a device to interact with it, users can control the device with simple hand motions [76].

Radar is a technology that has been used for both heart rate and respiratory rate detection simultaneously by many researchers [77,78,79,80,81,82] in particular, due to its demonstrated penetration properties. However, many times the breathing harmonics of a human are the same or close to the frequency of a heartbeat, so it is challenging to isolate the two signals and determine the heart rate and respiratory rate [83]. To avoid this conflict, Photoplethysmography (PPG), an optical technique, is an excellent choice to be used in conjunction with radar for non-contact heart rate determination [84].

#### 2.1.2. Photoplethysmography

Photoplethysmography (PPG) is an optical, noninvasive technique that has become common due to its high recorded accuracy and simplicity [85,86]. It has been used in smartphones and wrist-based wearables to estimate heart rate by using the flash of the device’s camera, a Light-Emitting Diode (LED), or Organic LED (OLED) due to its reported high accuracy when compared to an Electrocardiogram (ECC), which is considered gold-standard [87]. While plethysmography detects volume changes in the body [88], PPG utilizes a light source to detect volume changes inside a human’s blood vessel [89].

There are two modes of Photoplethysmography that are commonly used—PPG in transmission mode or reflectance mode. Both types utilize a light source to emit light into the tissue; in transmission mode, the light is sent through the tissue to a detector on the opposite side of the light source, while in reflectance mode, the light emitted is reflected, and its intensity is captured by the detector located next to the source [90]. This PPG mode is commonly used for wearables, predominately measured at the wrist, and PPG in transmission mode is commonly used by pulse oximeters when placed on a person’s finger [91]. As a light source, green light has been the preferred choice for wearables in reflectance mode due to its lower wavelength, avoiding interference with the blood vessels. For pulse oximeters in transmission mode, red has been the color of choice due to its penetration depth [92]—see Figure 5.

For both types of PPG, there are disadvantages. A too-tight fit of the device can decrease blood perfusion, with the increased pressure possibly causing decreased oscillations [93]; therefore, PPG in transmission mode is not ideal for continuous data capture, as the device also needs to stay attached to the body continuously. For the reflectance mode, the space between the skin and the devices’ sensors can cause significant noise due to a light incident caused by motion and too-loose fit [94].

#### 2.1.3. Remote Photoplethysmography

As the Pi-CON methodology promotes determining a patient’s heart rate passively, contactless, and continuously, PPG needs to achieve a heart rate reading from a distance. While PPG can be achieved via transmissive or reflectance PPG with a source emitting light onto a target and the detected difference in light intensity, remote PPG, or rPPG in short, uses a derivative of the reflectance mode by pointing a video RGB (Red, Green, Blue) camera to a target to record the difference of pixel density between heartbeats, with ambient light providing the light source [95].

Previous research has shown that the ideal Region Of Interest (ROI) for accurate results and heart rate extraction is the forehead due to easy accessibility, available algorithms to detect the forehead, and constant blood flow in the temporal artery [96,97]. In proper light conditions, limited motion, and close proximity to the light source, the video camera can detect the change in blood volume in the forehead’s blood vessels as the cardiac cycle goes through the systolic and diastolic phases (see Figure 6). With every heartbeat, the heart contracts in the systolic phase, the blood volume increases, and with that, the amount of light that is absorbed by the blood vessel [98]. This leads to subtle color changes in the forehead, which the camera can detect with minimal motion of the target under ideal light conditions [99]. While there are reported challenges applying this technology due to lighting conditions and patient motion [100,101], this concept has been showcased at the 2020 Tokio Olympics, where a camera system was installed 12 m away from athletes for successful non-contact heart rate monitoring [102].

#### 2.1.4. Thermal Camera

Detecting a person’s forehead temperature can be accomplished without touching a person by using a thermal camera, as it detects thermal radiation from the surface of an object [103]. Thermal imaging is a safe and noninvasive technology to detect differences in temperatures of objects in the line of sight [104], for which an infrared detector is added to a camera. This detector is used to make the light visible that is emitted by warm objects [105], which a human’s eye cannot detect.

The human skin Is an example of a warm object that radiates heat, and thermal imaging has been used to detect body temperature (see Figure 7), specifically after the onset of the COVID-19 pandemic with the attempt to contain the spread of the virus [106]. A human’s core temperature can be estimated by checking the temperature of a human’s skin on the forehead as this is more convenient and less time-consuming, especially for patients that do not cooperate with axillary, rectal, or oral methods [107]. The forehead lends itself well to this technology due to blood being a good indicator of the body’s core temperature, with the temporal artery running through the forehead. This blood vessel provides a constant blood flow and emits constant heat that can be captured conveniently since the blood vessel runs right below the surface of the skin [108,109].

Target distance, environmental noises, such as ambient temperatures, direct sunlight, or another source of heat, as well as head coverings (such as a headband) or skin condition (such as sweaty or dirty) can impact the temperature reading [110]. However, research has shown that thermal imaging systems provide accurate results for fever screening, including during the COVID-19 pandemic [111,112].

Table 1 analyses all discussed technologies and lists the advantages and disadvantages of each to develop a sensor based on the Pi-CON methodology [67,74,76,83,85,86,97,103,106,109,113,114,115,116]:

The Table 1 compares technologies that are commonly used for monitoring health conditions. Amongst those, radar, rPPG, and infrared stand out as favorable choices for Pi-CON due to: Non-invasiveness and safety: they do not require breaking the skin or causing any harm to the patient.Usability: they do not necessitate tight fitting accessories or markers on the patient’s clothing, improving comfort and reducing the risk of user error.Accuracy: these technologies are reported to be reliable and accurate in measuring vital signs, specifically when it comes to continuous data capture.Cost-effectiveness: they are generally inexpensive, making them accessible for wider use.

While the traditional PPG types offer valuable benefits, these technologies are not suitable for Pi-CON as they require sensors to be attached to the body. Radar, remote PPG, and infrared technologies, however, demonstrate significant promise for remote patient monitoring due to their combined advantages of patient comfort, reported accuracy, cost-effectiveness and continuous data generation possibilities. Radar’s ability to penetrate objects and function continuously makes it versatile for various settings. Infrared presents a valuable alternative for temperature monitoring without direct contact with the patient and also suits itself well for continuous data capture. Additionally, rPPG shares the advantage of continuous data collection, while not requiring markers on the body to detect heart beats.

It is, however, necessary to continue studying alternative sensing methods to advance this field and the application of the Pi-CON methodology.

## 3. Discussion

This work illustrates usability challenges and their impacts when home use medical devices or wearables are used in virtual health and introduces the Pi-CON methodology to eliminate these challenges and increase usability when a patient is taking their own exams in a virtual environment without a medical professional.

Pi-CON could be utilized while patients sleep or rest to avoid motion and retain the proper proximity to the sensor. Figure 8 highlights an example in a nursing home. By using permanently installed Pan–Tilt–Zoom (PTZ) cameras, the patients’ motion can be tracked, and health data can be monitored in near real-time by applying an integrated radar, rPPG, and infrared sensor as these patients are resting. The patient can be identified by applying the Viola–Jones algorithm stored in the Open-Source Computer Vision Library and isolates the forehead for the ROI [117]. PTZ cameras can monitor large areas due to the ability to rotate by 360°, allow zooming in on a patient’s forehead ROI, and adjust horizontally, as well as vertically while the patient to be tracked is inside the field of view [118].

As the data is acquired, it can be sent to a gateway device that will transmit the data to a IoT-enabled interface in near real-time, in which the patient, caregiver, or family members can track the patient’s health in near real-time. The user interface needs to be designed so illiterate users are encouraged to access the data and can interpret it correctly. Several PTZ cameras could be installed to track multiple individuals, with the gateway collecting all incoming data as well as video feeds and communicating this to a cloud-based interface so it’s accessible as needed.

Pi-CON could be expanded even further by using the same integrated sensors to track a human’s gait and falls, which has been accomplished before [112,113,114]. This would achieve the same as a wearable that can detect falls, however, without requiring the user to interact with the device. This sensor could detect falls and impacts (such as falls), in addition to the discussed vital signs, and notify a patient’s caregiver or family member in near real-time, potentially leading to help evaluate changes in a patient’s movement patterns after accidents or hospitalization and could predict falls by utilizing machine learning and a trained model.

## 4. Conclusions

The Pi-CON methodology described in this paper offers several key strengths that enhance the user experience and effectiveness of remote patient monitoring during virtual care.

Improved User Experience: Pi-CON eliminates the need for user interaction during data acquisition, reducing the burden on patients, especially those with limitations such as dexterity issues, cognitive decline, or visual impairments. This improves adherence and simplifies the process. It also focuses on non-contact sensing and avoids the discomfort and potential skin irritation associated with traditional wearables. This is particularly beneficial for long-term monitoring or patients with sensitive skin.

Enhanced Data Collection: the Pi-CON methodology enables the collection of health data continuously, providing a more comprehensive picture of a patient’s health compared to intermittent spot-check measurements. This allows for earlier detection of subtle changes in health indicators. Due to the non-contact sensing techniques, Pi-CON minimizes the need for direct patient contact, potentially reducing the risk of infection transmission in clinical settings.

In addition to these listed benefits, there are also limitations, specifically when it comes to the performance of the sensors establishing health parameters remotely without user intervention. It is therefore recommended to initially apply the Pi-CON methodology in an experiment to remotely monitoring patients while in resting condition to avoid motion artifacts. Additionally, the patient preference for an omnipresent and ubiquitous sensor compared to the preference and usability of a PGHD device also needs to be verified.

As an outlook, Pi-CON should be utilized as a framework as new PGHD devices are developed. Pi-CON should also guide future research to increase data accuracy and usability for wearables and home use medical devices when a medical professional is not present to observe or perform a medical exam. The identified challenges in RPM, such as device setup, the need for patients to actively initiate and position the device, or to have something attached to the patient’s body can introduce errors and limits user motivation to use the device, affecting the accuracy of measurements. The Pi-CON methodology addresses these challenges and minimizes the burden on patients as it prioritizes continuous monitoring and explores non-contact technologies, without the need to set up a device or attach a sensor to a patient body to avoid can discomfort or limited mobility.

While there are sensor recommendations in this paper on how to apply Pi-CON, the methodology is not limited to a specific sensor. It can be applied using various sensor technologies like radar, remote PPG, and infrared, offering flexibility based on specific monitoring needs. This encourages ongoing research and development to explore new sensors and advanced signal processing techniques to apply Pi-CON, paving the way for future improvements in accuracy, functionality, and data management.

In addition to the very limited user interaction, the patient can also benefit from the Pi-CON methodology when the ubiquitous sensor is IoT-enabled so that generated data is visible in real-time in a cloud-based interface. This would allow family members, friends, or a healthcare provider to follow the patient’s current heart rate, respiratory rate, and body temperature in near real-time on their device or on a website to observe the patient’s current health status without being in the same vicinity. Since little to no human interaction is required by the patient, access to live data could also help prevent false alerts when patients accidentally activate a medical alarm.

One iteration of the proposed sensors could be driven by the need to acquire additional health data. This could, for instance, include passive, continuous, and non-contact blood pressure estimation with a non-invasive estimation called Pulse Transit Time, or PTT, which describes the travel duration of a pulse wave from one arterial site to another, with the speed being directly proportional to blood pressure [119].

The Pi-CON methodology represents a shift towards more user-centric health and comprehensive remote patient monitoring in virtual health. It prioritizes patient comfort, data accuracy, and infection control, ultimately aiming to improve healthcare outcomes by facilitating early detection of health concerns and better chronic disease management.

## Figures and Tables

**Figure 2 sensors-24-02260-f002:**
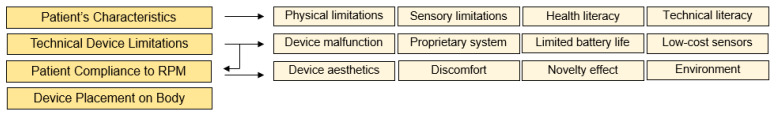
RPM usability impact model.

**Figure 3 sensors-24-02260-f003:**
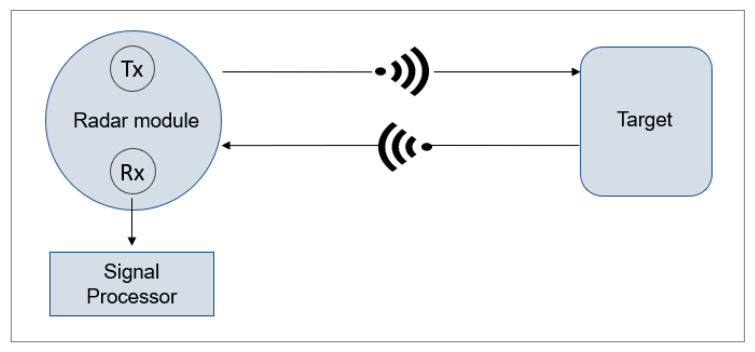
Radar functionality.

**Figure 4 sensors-24-02260-f004:**
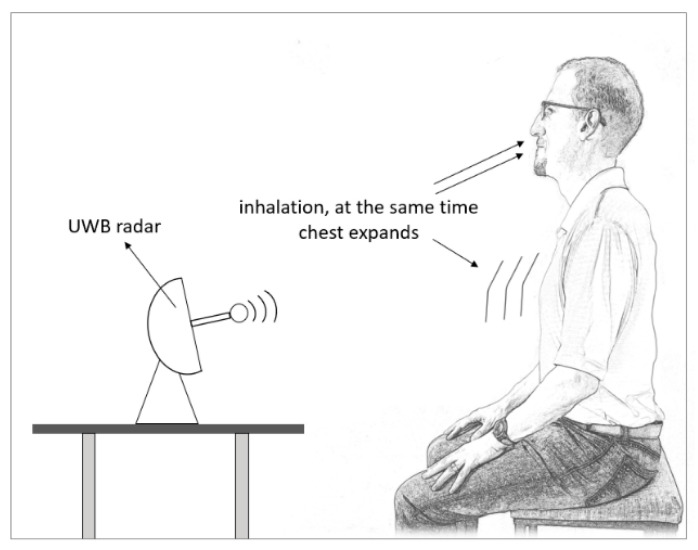
UWB radar is used to estimate respiratory rate.

**Figure 5 sensors-24-02260-f005:**
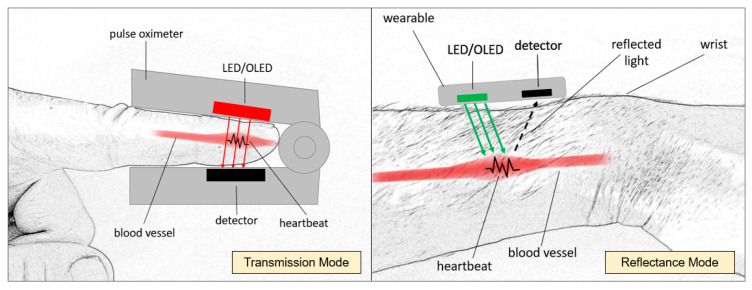
PPG illustrating transmission mode (**left**) and reflectance mode (**right**).

**Figure 6 sensors-24-02260-f006:**
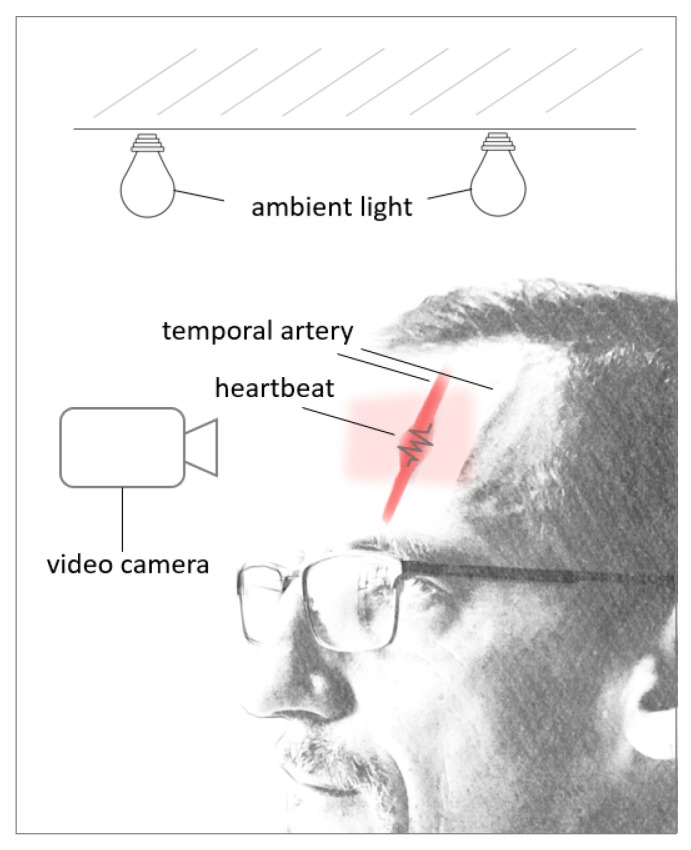
rPPG during systolic phase.

**Figure 7 sensors-24-02260-f007:**
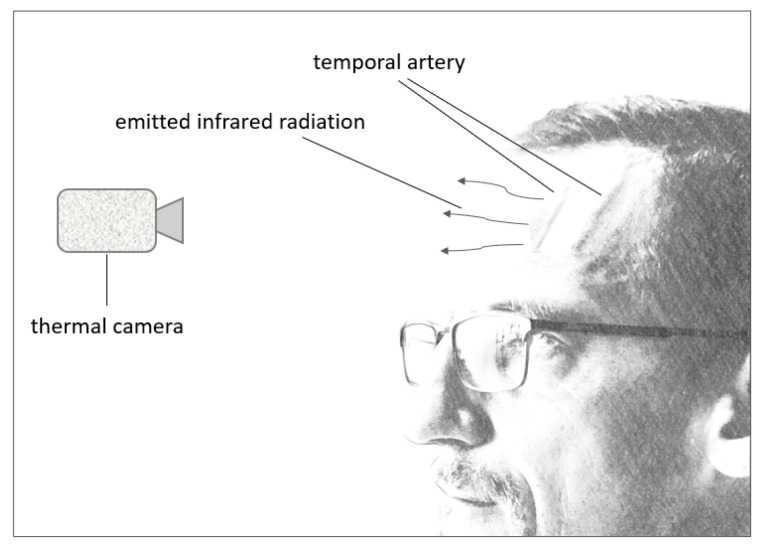
Thermal camera functionality.

**Figure 8 sensors-24-02260-f008:**
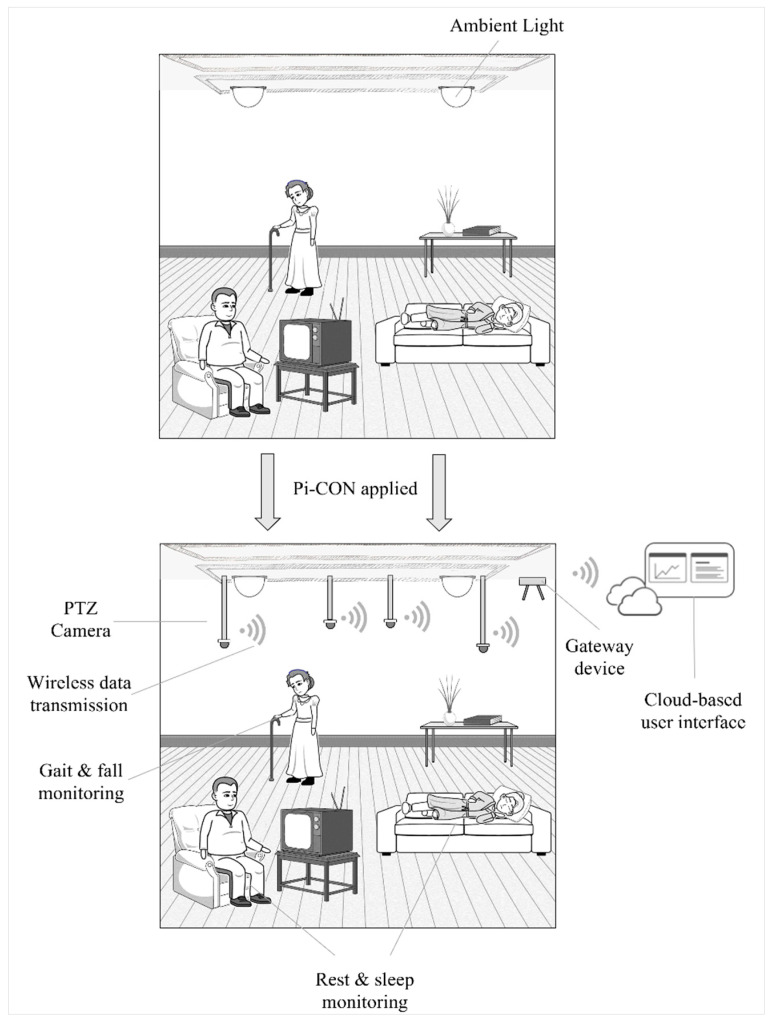
The Pi-CON methodology applied in a nursing home.

**Table 1 sensors-24-02260-t001:** Pi-CON Sensor Analysis.

	Advantages	Disadvantages	Health Parameter
Photoplethysmography—Transmission mode	- noninvasive - safe - reported reliability and accuracy - inexpensive - no need to wear markers in clothing	- too-tight fit can cause decreased oscillations and therefore inaccurate results - should be worn for spot-checks only - requires to be attached to body	- Blood oxygen saturation, heart rate (not ideal if continuous)
Photoplethysmography—Reflectance Mode	- noninvasive - safe - reported reliability and accuracy - inexpensive - no need to wear markers in clothing - well suited for continuous data generation	- space between skin and sensor - can cause noise and therefore - inaccurate results - requires to be attached to body - operation reliant on battery life	- Blood oxygen saturation heart rate, sleep quality
Remote Photoplethysmography	- noninvasive - safe - inexpensive - no need to wear markers in clothing	- good light conditions needed - too much motion will lead to inaccurate results - subject needs to be in line of - Sight - close proximity to sensor	- Heart rate, sleep quality
Radar	- noninvasive - safe - reported reliability and accuracy - inexpensive - no need to wear markers in clothing - well suited for continuous data generation - ability to penetrate through objects and walls	- need to choose right radar - isolating hear rate from respiratory rate	- Respiratory rate, heart rate, sleep quality, gait, fall and motion
Infrared	- noninvasive - safe - inexpensive sensors - valid body temperature alternative to axillary, rectal, or oral methods - no need to wear markers in - clothing	- subject needs to be in line of Sight - close proximity to sensor - temperature reading can be impacted by o environmental noises o head coverings sweaty skin	- Body temperature

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
