# Peer review of "Introducing the Pi-CON Methodology to Overcome Usability Deficits during Remote Patient Monitoring"

_sensors, 2024, doi:10.3390/s24072260_

Round 1

Reviewer 1 Report

Comments and Suggestions for Authors

The authors first describe telehealth and its subcomponents, focusing on Remote Patient Monitoring (RPM), and present four aspects that impact measurement devices' usability. Then they review some technologies aimed at supporting (almost) passive, non-contact, and (almost) continuous reading of a very limited number of vital signs (namely body temperature, heart rate, and breathing rate), in an attempt to overcome the highlighted usability limitations. Among the analyzed technologies, UWB radar, infrared, and remotePPG are the suggested ones for the implementation of the proposed Pi-CON methodology. Finally, a very brief description of a potential use-case is provided.    

It is unclear to this reviewer what the rationale of the work is, and how the results presented can "be utilized as a model by PGHD manufacturers as new PGHD devices are defined and guide future research to increase data accuracy and usability for wearables and home use medical devices – line 499-502".

All reviewed technologies, and also the selected ones, have important disadvantages (table 1). In the reviewer's opinion, these partially affect usability, certainly affecting data accuracy (e.g. environmental conditions, movements, etc.) and definitely impacting applicability in real scenarios. A deep discussion of these aspects and how to mitigate such impacts to achieve a reliable application of the aforementioned technologies is required to understand the proposed methodology.

In addition to the previous concerns, hereafter some other points and questions that could help to improve the work:

Is the RPM Usability Model shown in Figure 2 proposed by the authors or coming from the literature?

Section 1.4: is the spirometer “early termination” the only technical device limitation worthy of mention, also considering that spirometry is not included in the set of vital parameters supported in Pi-CON?

Section 1.6: same as above with respect to device body placement and blood pressure.

Line 269: are 400.000 in-hospital deaths per day correct? Probably per year.

Section 2.1.1: Is there in the literature evidence that such technologies is able to measure 4-12 mm distances (chest displacement)? 1 How deals with subject’s movement in measuring respiration rate exploiting UWB?

Section 2.1.3: How to ensure proper readings in non-optimal environmental conditions, which can happen quite frequently in settings other the laboratory experiments? What is the range of distances to ensure an accurate reading of heart rate?

Section 2.1.4: same questions as above with respect to thermal camera and body temperature. In addition, has the degree of acceptance of the use of video cameras and the perceived impact on the privacy of the subjects been assessed?

Section 3: how Pi-CON manages many subjects on the scene?

Finally, the paper does not follow the journal template. Reference style and citing style are not correct. Not all references are cited in the text (e.g. Staderini 2002). Some terms are used in different forms (home use vs home-use).

Overall, the authors need to clarify the intended scope of the work and provide additional details to help the reader in understanding the level of applicability of the proposed Pi-CON in real scenarios. 

Comments on the Quality of English Language

The quality of English is good  

Author Response

Thank you for your comments; please see the attachment.

Reviewer 2 Report

Comments and Suggestions for Authors

The manuscript introduces the Pi-CON methodology, describes how to overcome the challenges, and provides user-friendly and intuitive devices to determine vital signs. Some sensors have been chosen based on their ability to continuously capture data hands-free without touching a patient. Although the work submitted by the authors is well-written and detailed, here are some suggestions for improvement:

- The abstract could contain, in the final part, more information about the final version of the Pi-CON. For example, inserting more information about the communication between the sensor, gateway, and system to acquire this data can help the readers. It's just an example; maybe the authors can elaborate more on the possible ideas to improve the abstract.

- The keywords are straightforward. I suggest adding more terms that are focused on the work.

- I suggest inserting some text between Section 1 and Subsection 1.1.

- MDPI does not recommend the way that the authors are using to cite related works. Only numbers are allowed throughout the text, and the list of references must contain the respective numbers. Please check it: https://mdpi-res.com/data/mdpi_references_guide_v5.pdf

- All acronyms need to be revised to the following standard, for example:

patient-generated health data (PGHD) > Patient-Generated Health Data (PGHD)

- Line 269: why is it in blue?

- The main question is: Why not explore the Internet of Things issues more? After reading the whole manuscript, I felt a lack of usage of Internet of Things terms. For example, from the idea illustrated in Figure 8, it is clear that the whole system is based on concepts from the Internet of Things. However, the authors only mention this term two or three times. Also, after proposing a solution using the Pi-CON, the work could elaborate on the next steps or discuss some ideas for implementing it. In this sense, the authors might discuss some concepts about the data flowing from the data acquisition to the visualization.

Best regards.

Author Response

(The authors gave the same response as above.)

Reviewer 3 Report

Comments and Suggestions for Authors

The abstract does not convey any potential problem and the discussion related to the contribution of the paper. The objective and the methodology of the paper need to be covered. The main findings of the study also need to be highlighted.

The paper is categorized as review article. The authors state in line 93 that “a novel framework for gathering RPM data, is proposed. The title seems like a research article. The title and other parts of the paper need to be consistent.

Section 2 presents details of the technologies that may be suitable for remotely and continuously monitoring patients. It is not clear which technology is used in the proposed method. The details of the implementation are missing. No experimental section and details are presented. The discussion includes an example however the implementation mechanism and other details are missing.

Author Response

(The authors gave the same response as above.)

Round 2

Reviewer 1 Report

Comments and Suggestions for Authors

According to this reviewer, the quality of the paper has been improved during the revision process, making the scope of the work more understandable. Most of the points highlighted in the previous step have been sufficiently addressed by the authors: a wider presentation of technical device limitations has been added as well as references supporting the applicability of RemotePPG; conclusions have been reformulated and made more clear; most of the potential challenges have been introduced.

Hereafter are some improvement points:

In section 2.1.1, the reference provided in the reply letter should be added to answer the review question.

As done for remotePPG in section 2.1.3, it could be interesting to discuss (and support with references) how the challenges regarding thermal cameras described in section 2.1.4 have been addressed in the state-of-the-art.

Table 1 should be improved by adding a new column reporting the vital parameter(s) associated with each of the discussed technologies.

Authors are finally invited to report the main limitation of the proposed methodology, which in the reviewer’s opinion is the applicability of Pi-CON only in resting conditions of subjects, considering the available/selected technologies.   

Re-reading is recommended for the removal of typos such as in lines 488, 492, and 545. 

Comments on the Quality of English Language

English is good. 
